# Modified Graphene Oxide-Incorporated Thin-Film Composite Hollow Fiber Membranes through Interface Polymerization on Hydrophilic Substrate for CO_2_ Separation

**DOI:** 10.3390/membranes11090650

**Published:** 2021-08-25

**Authors:** Ook Choi, Iqubal Hossain, Insu Jeong, Chul-Ho Park, Yeonho Kim, Tae-Hyun Kim

**Affiliations:** 1Research Institute of Basic Sciences, Incheon National University, 119 Academy-ro, Yeonsu-gu, Incheon 22012, Korea; ooksclub@inu.ac.kr (O.C.); iqubal.chem.ru.08@gmail.com (I.H.); jis_3088@naver.com (I.J.); yeonho@inu.ac.kr (Y.K.); 2Organic Material Synthesis Laboratory, Department of Chemistry, Incheon National University, 119 Academy-ro, Yeonsu-gu, Incheon 22012, Korea; 3Jeju Global Research Center (JGRC), Korea Institute of Energy Research (KIER), 200 Haemajihaean-ro, Gujwa-eup 63357, Jeju Specific Self-Governing Province, Korea; chpark@kier.re.kr

**Keywords:** hollow fiber membrane, interfacial polymerization, modified graphene oxide, CO_2_ separation, composite mixed matrix membranes

## Abstract

Thin-film composite mixed matrix membranes (CMMMs) were fabricated using interfacial polymerization to achieve high permeance and selectivity for CO_2_ separation. This study revealed the role of substrate properties on performance, which are not typically considered important. In order to enhance the affinity between the substrate and the coating solution during interfacial polymerization and increase the selectivity of CO_2_, a mixture of polyethylene glycol (PEG) and dopamine (DOPA) was subjected to a spinning process. Then, the surface of the substrate was subjected to interfacial polymerization using polyethyleneimine (PEI), trimesoyl chloride (TMC), and sodium dodecyl sulfate (SDS). The effect of adding SDS as a surfactant on the structure and gas permeation properties of the fabricated membranes was examined. Thin-film composite hollow fiber membranes containing modified graphene oxide (mGO) were fabricated, and their characteristics were analyzed. The membranes exhibited very promising separation performance, with CO_2_ permeance of 73 GPU and CO_2_/N_2_ selectivity of 60. From the design of a membrane substrate for separating CO_2_, the CMMMs hollow fiber membrane was optimized using the active layer and mGO nanoparticles through interfacial polymerization.

## 1. Introduction

Carbon dioxide (CO_2_) separation or capture is highly desired in order to contribute to solving, or at least reduce, the worldwide threat of global warming brought about by the emission of greenhouse gases (GHGs) such as carbon dioxide (CO_2_), methane (CH_4_), nitrous oxide (N_2_O), hydrofluorocarbons (HFCs), perfluorocarbons (PFCs), and sulfur hexafluoride (SF6) [1,2]. These GHGs are continuously released into the atmosphere from the burning of fossil fuel sources to fulfill the unavoidable energy demand of modern civilization throughout the world [1]. To mitigate energy generation-related CO_2_ emissions, the Intergovernmental Panel on Climate Change (IPCC) stated in its recent assessment report that CO_2_ release to air must be substantially reduced to achieve stabilization of the atmospheric CO_2_ concentration during the 21st century [1,3].

Carbon capture and storage (CCS) is a technology that can capture, purify, compress, and transport CO_2_ and store it safely under the Earth’s surface [4,5,6,7]. There are several types of CO_2_ separation or capture approaches depending on various parameters, such as concentration of CO_2_ in the gas stream, the pressure of the gas stream, and the fuel type (solid or gas) for selecting the capture system [8,9].

Membrane-based separation technology has attracted significant attention since the first use of membrane technology for small-scale gas separation in the late 1970s, due to its intrinsic advantages over the conventional technologies (e.g., amine scrubbing, sorbent adsorption, or cryogenic distillation) [10,11]. This environmentally friendly technology ensures potentially less energy consumption, lack of mechanical complexity, and its modularity allows easy scale-up [12]. Hence, membrane gas separation allows for more straightforward system operation and lower capital and processing costs and can be accomplished with smaller footprints than chemical absorption [1,5].

A hollow fiber membrane provides intrinsic advantages such, as higher packing density and a larger membrane area per unit volume, a reduction in module production cost, and easier operation and high gas permeation than a flat sheet membrane and tubular membrane processes [13,14,15]. For these highly desired reasons, the hollow fiber process is widely used in actual separation operations for industrial-scale applications. However, irrespective of the form of the membrane modules used in gas separation application, the potential applications of a given membrane technology largely depend on the ability of the membrane material to exhibit high separation performance. Unfortunately, pure polymer-based membrane modules for gas separation always follow a trade-off relationship between gas permeability and selectivity [11,16,17]. Briefly, a high permeable membrane shows low selectivity and vice versa. This widely known performance limitation is called the “Robeson Upper bound”, and reduces the commercialization potential of membrane technology despite the benefits of low cost and easy scalability [18,19].

Numerous approaches have been reported on developing high-performance membrane materials to overcome this trade-off relationship. Among them, hybrid membranes (e.g., mixed-matrix membranes MMMs or composite mixed-matrix membrane CMMMs) have been recognized as a state-of-the-art way to combine the advantages of a polymer (mechanical flexibility, facile processing, low cost, and easy to scale up) with inorganic fillers (excellent thermo-mechanical stability, high free volume, and selectivity) or incorporating reactivity-selective CO_2_-carrier into the polymer matrices to facilitate the transport of CO_2_ [15,20,21,22].

Nonetheless, most of these developed materials have usually been reported as self-standing thick films, typically in the order of 40–100 microns [15]. However, the successful development of membrane materials is benchmarked with the improvements in CO_2_ flux (permeance), which can be achieved by the fabrication of thin-film composite (TFC) in the form of a flat sheet, or hollow fiber membrane with a selective layer, typically with a thickness of few hundred nanometers [15]. The hollow fiber module is more advantageous due to the compact membrane area and lower production cost required to achieve the targeted separation performance [22].

However, the fabrication of the ultrathin composite layer on the hollow fiber is very challenging, and it is almost impossible to maintain the original permeance of the material evaluated as self-standing thick films. Additionally, when the selective layer comprises a hybrid material, two distinct phases increase the complexity to achieve a defects-free coating. Furthermore, owing to the hollow fiber configuration, the coating procedure needs further optimization due to its curved topology [15]. For these reasons, limited studies have been reported on fabricating hollow fiber thin composite membranes with a hybrid selective layer for CO_2_ separation applications.

Another quite promising potential for highly efficient gas separation is to use fixed carrier membranes comprising carriers [14,23,24,25] These have been demonstrated to be very effective in simultaneously improving gas permeability and selectivity through the reversible reactions between reactive carriers and the targeted gas CO_2_ [23,24]. Most fixed carrier membranes reported in the literature contain amine moieties as the CO_2_-reactive functional groups [25]. These membranes generally present CO_2_ permeance from 100 to 6000 GPU and CO_2_/N_2_ selectivity above 50 [24].

However, excessive amine-containing membranes, such as primary amine-based polymer membrane, poly(vinylamine) (PVAm) often have high crystallinity due to the strong intermolecular interaction from the hydrogen atoms [25]. The high crystallinity not only reduces the membrane’s separation performance, but also makes the membrane brittle. In contrast, secondary or/and tertiary amines containing a polymer have the weaker intermolecular interaction, and thus show low crystallinity. Moreover, the amine groups may be oxidized by the oxidant (mainly O_2_), or the acid gas (mainly SO_2_) in the flue gas may also react with the amine groups. Ultimately, the separation performance of the membrane may deteriorate [24].

To solve those limitations of a facilitated transport membrane, incorporating an inorganic nanofiller, such as modified graphene oxide (mGO), could be an effective strategy for CO_2_ separation [15,22]. Graphene oxide (GO) has been successfully used in many studies to improve the gas separation performance of resultant membranes due to its high aspect ratio, layered structure, which can provide gas transport channels with different path tortuosity for different gases [15]. Moreover, GO provides improved thermal and mechanical properties as well [26]. However, hybrid membranes containing GO filler are rarely reported in the form of the thin composite selective layer in the hollow fiber module.

In this work, we developed fixed carrier thin-film composite mixed matrix membranes (CMMMs) prepared by the interfacial polymerization (IP) using modified GO (mGO) filler on the hollow fiber support. The hollow support fiber was prepared by adding and mixing polydopamine (PDA) and poly(ethylene glycol) (PEG) into poly(ether sulfone) (PES) to promote pore formation in the substrate and enhance its hydrophilicity, thereby increasing the resultant permeate flux and the binding force the thin-film composite layer and the PES support layer during interfacial polymerization. The composition of the thin-film composite layer and support layer of CMMM is presented in Figure 1.

Interface polymerization (IP) has been used in this present study as it is a well-established method for preparing thin-film composite membranes [22,23,24,25,27]. The IP process has several advantages, such as the formation of ultrathin active thin film, a diversity of monomers, the tunable functional groups, the minimization of defects inside the thin films, and lack of strict requirements for reactant purity [3,23,25,27]. Ultrathin membranes made by IP offer high permeance and provide adhesion between the IP layer and the support due to the penetration of the IP layer into the porous support, and the IP technique can be easily scaled up to an industrial scale [23,24]. Sodium dodecyl sulfate (SDS) surfactant is used to enhance the reactivity of reactant in the IP method and thereafter to enhance the permeance and selectivity of the resultant thin-film composite layer. PEI monomer of IP is used as a fixed carrier for selective CO_2_ transport, while mGO that is incorporated during the IP method is expected to be intercalated inside of the thin film to enhance the free volume IP polymer, and thereby form gas transport pathways through the thin-film composite layer in the CMMMs.

The fabricated mGO thin-film composite hollow fiber membranes (CMMMs) were both chemically and physically analyzed. The effect of hydrophilicity in the support layer, the effect of SDS concentration, and the effect of mGO-loading in the thin-film composite layer on the morphology and physicochemical properties of the resultant CMMMs was investigated. Finally, those effects (hydrophilicity, SDS, and mGO loading) were also examined against the gas permeation properties using CO_2_ and N_2_.

## 2. Experimental

### 2.1. Preparation of PEG and PDA-Incorporated Poly(ether sulfone) Hollow Fiber Membrane Substrate

The hollow fiber membrane as a substrate was prepared by the spinning process using poly(ether sulfone) (PES) flakes (Ultrason^®^ E 6020 P, BASF, Ludwigshafen, Germany), polyethylene glycol (PEG 6000, Sigma-Aldrich, St. Louis, MI, USA), and dopamine and ammonium persulfate (Sigma-Aldrich, St. Louis, MI, USA). The mixed polymers were then subjected to agitation at 60 °C and a speed of 160 rpm for three days in a tank, with N-methyl pyrrolidone (SAMCHUM CHEMICALS, Seoul, Korea) added as a solvent, to fabricate a dope solution, as shown in Table 1 and Appendix A. Following agitation, a vacuum pump removed small bubbles from the dope solution for 24 h before starting the spinning process.

The spinning process was performed as follows. First, the polymer dope solution was delivered to a spinneret (0.15/0.9) using a gear pump. In the process, deionized water (DI water) was used as an internal coagulant, and the HPLC pump (Series pump, Lab Alliance, Syracuse, NY, USA) was employed. After going through the spinneret, the dope solution underwent phase inversion in a coagulation bath and was then washed in a secondary coagulation bath and subsequently wound in a winder. While wound on a bobbin, the fabricated hollow fiber membrane was washed with running water at 40 °C for a week to remove all remaining solvents, and then subjected to post-treatment using methanol. After the post-treatment process, the membrane was dried at room temperature.

### 2.2. Preparation of Thin-Film Composite Membranes with Modified Graphene Oxides

Modified graphene oxide (mGO) was synthesized using graphene oxide (GO) as follows: 0.6 g of GO was added to 400 mL of deionized (DI) water and then dispersed using ultrasonication. Subsequently, 4 g of sodium hydroxide (SAMCHUM CHEMICALS, Seoul, Korea) was added, and the solution was subjected to magnetic stirring for 1 h. Afterward, 10 mL of hydrochloric acid (SAMCHUM CHEMICALS, Seoul, Korea) was added to 400 mL of DI water. This solution was then added to the previously obtained mixture, then stirring and centrifuge was employed to allow a reaction to occur for 1 h. Subsequently, the mixture was washed with DI water several times until the pH reached a neutral level, and the product was dried in a vacuum oven at 80 °C to obtain mGO.

The spun hollow fiber membrane and the synthesized mGO were subjected to interfacial polymerization to fabricate thin-film composite hollow fiber membranes. The aqueous solution for interfacial polymerization was prepared by adding polyethyleneimine (PEI, branched, Sigma-Aldrich, St. Louis, MI, USA), an amine-based monomer, to DI water while also adding varying concentrations of sodium dodecyl sulfate (SDS, Sigma-Aldrich, St. Louis, MI, USA) and mGO. The organic solution for interfacial polymerization was prepared by mixing and stirring trimesoyl chloride (TMC, Sigma-Aldrich, St. Louis, MI, USA), 1,3,5-benzenetricarbonyl trichloride), an acyl chloride-based monomer, and n-hexane (99.9%, Doosan, Seoul, Korea) into a solution.

Each fabricated solution was coated on the inside of the modulated hollow fiber membranes using a syringe pump. The coating process was performed as follows. First, the prepared aqueous solution was injected for one minute, then a purge by nitrogen gas was performed for one minute. After this step, the organic solution was injected for one minute, again followed by one minute of nitrogen gas purging. The obtained polyimide film was then subjected to stabilization at room temperature for one hour and subsequently heat-treated in an oven at 80 °C for ten minutes. Afterward, the film was washed with DI water and dried. This procedure was repeated several times to remove all remaining solvents. The coating process was performed at the same reaction temperature and with the same reaction time for each specimen to make this interfacial polymerization process more reproducible. The compositions of each monomer and the ratios of SDS and mGO are summarized in Table 2.

### 2.3. Characterization of mGO and Thin-Film Composite Hollow Fiber Membranes

The characteristics of the hollow fiber membrane substrate fabricated by the spinning process, the synthesized mGO, and the thin-film composite hollow fiber membranes were analyzed as follows. An X-ray diffractometer (XRD, Rigaku SmartLab, Tokyo, Japan) was employed to analyze changes in the d-spacing of the GO and mGO within the range of 5° ≤ 2θ ≤ 40°. The morphological analysis was performed using a transmission electron microscope (TEM, Talos F200X, Thermo Fisher Scientific, Waltham, MA, USA). Raman spectroscopy (Renishaw, inVia Qontor, Gloucester, UK) was used to measure the D/G bandgap and determine whether the intended phases had been synthesized. X-ray photoelectron spectroscopy (XPS, PHI 5000 Versaprobe II, Chanhassen, MN, USA) and attenuated total reflectance Fourier transform infrared spectroscope (ATR-FTIR) were also employed to analyze in the wavenumber range 600–4000 cm^−1^. the morphological and chemical characteristics of the thin-film composite membranes and the effect of the mGO content on them. The cross-section and surface of the hollow fiber membranes and the substrate were observed and analyzed using a field emission-scanning electron microscope (FE-SEM, JEOL/JSM_7800F, Tokyo, Japan) In order to obtain an image of the cross-section of the hollow fiber membrane, it was cut using liquid nitrogen after wetting with DI water. The surface roughness of the membranes was examined using an atomic force microscope using non-contact tapping mode (AFM, Bruker, MULTIMODE-8-AM, Billerica, MA, USA). The surface hydrophilicity of the fabricated hollow fiber membranes was analyzed by measuring contact angles (Phoenix 300 Plus, SEO, Seoul, Korea). It was measured at room temperature by the sessile-drop method using 3 μL of DI water. Each sample was measured at 3 or more different locations and the average value was used.

### 2.4. Measurement of Gas (CO_2_, N_2_) Separation Performance

The single-gas permeance of the mGO-thin-film composite hollow fiber membranes was measured using an air circulation oven at 25 °C and an operating pressure range of 0.25–2.0 bar. The permeance was measured while varying the pressure level (Table 3 and Appendix A). First, the volume of the gas that permeated through hollow fiber membrane modules was measured using a bubble flow meter (Holiba VP-1, Kyoto, Japan). Then, the measured volume of each gas (CO_2_ and N_2_) was converted into permeance using the equation shown below.
(1)$$P=\frac{Q_{P}}{A\;\times \;\Delta P}$$

Here, Q_P_ refers to the permeate flow rate of the gas permeating through membrane modules, ΔP is the gas pressure difference across the membrane, and A is the effective area of the membrane. Under the SI system, membrane permeance is expressed as mol/(m^2^ s Pa) or cm^3^ ((STP)/cm^2^ cmHg sec) by convention. However, for hollow fiber membranes, in particular, gas permeation units (GPU) are preferred and more widely used, where 1 GPU = 1 × 10^−6^ cm^3^ ((STP)/cm^2^ cmHg·s) [28]. The ideal selectivity of the single-gas permeation (pure gas; CO_2_; or N_2_) can be expressed as Equation (2) below.
(2)$$a_{i,j}=p_{i}/p_{j}$$

## 3. Results and Discussion

### 3.1. Synthesis and Characterization of Modified GO (mGO)

As mentioned above, the present study aimed to develop mGO-thin-film composite hollow fiber membranes using mGO to improve gas permeance and selectivity. The mGO was prepared by treating GO with NaOH and HCl to reduce the oxygen groups it contained.

The synthesized mGO was analyzed as follows. First, its microstructure was examined with TEM images. As shown in Figure 2a, GO has a structure composed of flat and soft nanosheets, while mGO contains irregularly shaped nanosized particles 5–20 nm across (Figure 2b). Brunauer–Emmett–Teller analysis (BET) was employed to determine the gas sorption properties of the GO and mGO in more detail (Figure 2c). The results showed that the sorption amount of the mGO was about twice as large as that of GO. In addition, the pore size distribution curve also indicates that the pore size increased from 19.8 nm (for GO) to 28.4 nm (for mGO) (Appendix A). These results indicate that the GO has been successfully modified in our experiments.

The microstructure of the mGO was further analyzed using XRD (X-ray diffraction), as shown in Figure 2d. There was a difference in the peak position resulting from changes in d-spacing: GO (0.74 nm, 11.98°) and mGO (0.70 nm, 12.72°), indicating that mGO has maintained its ordered stacking sequence with only a slightly reduced distance [29].

Fourier-transform infrared (FTIR) spectroscopy was employed to observe changes in the oxygen-containing functional groups contained in the GO and mGO (Figure 2e). A stretching peak attributed to carbonyl C=O was found at 1729 cm^−1^ in both the GO and mGO, and another stretching peak attributed to epoxide C–O was observed at 1055 cm^−1^. However, the C–OH bending peak at 1403 cm^−1^ in the mGO was lower in intensity than the equivalent peak in GO [30,31].

The structure of the mGO was further analyzed using Raman spectroscopy. This technique is widely used when analyzing defects such as edges, vacancies, or ripples, or disorders observed in graphite or graphene oxide [32]. The D band was found at 1341 and 1335 cm^−1^ for GO and mGO, respectively, while the G band with a sharp peak shape was observed at 1580 and 1577 cm^−1^ for GO and mGO, respectively (Figure 2f). In the Raman spectra, the G band is a commonly observed primary Raman scattering spectrum attributed to the C–C bond strengthening by sp^2^ carbon atoms [33]. This indicates that the increased number of sp^2^ carbon atoms in the mGO shifted their G band to the lower wavenumber side, causing the D band to decrease in intensity. In addition, the ID/IG ratio after the reaction was measured as 0.866, which indicated a reduction in the number of oxygen-containing functional groups in mGO [34].

The presence of defects in mGO was thus confirmed. Furthermore, the defected mGO is expected to increase the specific surface area and expand the pathway for gas molecule diffusion [35,36].

### 3.2. Preparation and Characterization of mGO-Thin-Film Composite Mixed Matrix Membranes (CMMMs)

A series of CMMMs was prepared by IP method using organic phase TMC and aqueous phase PEI reactant in the presence of mGO-filler and SDS surfactant on the hydrophilic hollow fiber substrate (using PES + PEG +PDA mixed substrate) as described in the experimental section and illustrated in Figure 1. That schematic diagram (Figure 1) represents the detailed structure and components of the thin-film composite hollow fiber membrane developed in the present study. In addition, some reference membrane without mGO and/or without SDS was also prepared for comparison.

The structures of the PES substrate (fabricated by the spinning process) and the thin-film composite layers, i.e., PT, PTS, and PTSM-based composite hollow fiber membranes, were further analyzed using ATR-FTIR (Figure 3). First, some peaks resulting from the PES substrate were found: one peak was attributed to O=S=O symmetric stretching at 1150 cm^−1^, and two peaks were attributed to aromatic C=C stretching at 1580 and 1484 cm^−1^ [37,38]. It was also found that in the substrates that contained PEG and DOPA, a C-H stretching vibration peak resulting from PEG at 2875 cm^−1^ and a broad N-H and -OH peak resulting from DOPA at 3300–3700 cm^−1^ increased in intensity [39].

In the thin-film composite hollow fiber membranes after interfacial polymerization, a –CONH- peak was observed at 1652 cm^−1^, attributed to the polyamide layers formed via crosslinking between the PEI and TMC [22]. Meanwhile, an –OH peak at 3500 cm^−1^, a characteristic peak of SDS, had a relatively higher intensity than the original PES. This peak confirmed that the intended layers had been formed in the PT, PTS, and PTSM membranes [40,41]. However, such a difference could not be determined between the PTS and PTSM specimens (containing mGO), which may be due to the overlap of these two corresponding IR peaks. To this end, XPS analysis was performed (Figure 4).

The high-resolution C1s region (282–292 eV) of the PTSM specimens with PTS and mGO was analyzed by XPS analysis. In both specimens, the C-C and C-H peaks associated with carbide species were observed at the same position, that is, at 284.5 eV, as shown in Figure 4a,b [42]. In PTSP, a peak related to the C=O bonds formed after interfacial polymerization was found at 285.8 eV, and a peak resulting from the amide OCN was observed at 286.7 eV. A carboxyl group (O=C–O) peak found at 291.0 eV, attributed to GO, had a higher intensity (Figure 4b) [43,44,45].

These results confirmed the successful fabrication of both hydrophilic hollow fiber membranes by the spinning process and thin-film composite hollow fiber-based composite mixed-matrix membranes that contained SDS and mGO.

The surface morphology of the hydrophilic substrate was further analyzed using SEM and contact angle measurement (Figure 5). Figure 5a,b show the cross-section and surface images of the spun PES substrate. These images show that the outside of the membrane is denser, and the finger-like structure becomes more pronounced on the inside. Additionally, some pores can be observed on the surface. The PES substrate with this porous structure, shown in Figure 5b, was obtained as during the hollow fiber membrane spinning process, the coagulant bath was kept at 40 °C above room temperature to ensure a high permeate flux. The specimen was subjected to rapid solvent exchange in the subsequent washing process while being washed with running water at 60 °C. The addition of PEG and DOPA (dopamine) to the PES substrate also made the resultant hollow fiber membrane hydrophilic with a contact angle of about 52° (Figure 5c). Indeed, a contact angle of 54° could be achieved by adjusting the DOPA concentration (0.005–0.1 wt.% in the dope solution) (Appendix A). In the process, the optimum DOPA concentration needed to fabricate a hydrophilic substrate was determined.

Next, the microstructure of the thin-film composite hollow fiber membranes was analyzed using FE-SEM (Figure 6). Figure 6a, e show surface and cross-section images of the hollow fibers. Active layers formed via the crosslinking reaction between PEI and TMC are visible. Figure 6b–d,f–h show the effect of the SDS concentration, added as a surfactant for interfacial polymerization, on the surface and layer thickness. Increasing the surfactant concentration is known to increase the roughness of the membrane surface and the thickness of its active layers. The result indicates that the presence of SDS significantly enhances the reactivity of interface polymerization. The present study also showed similar results, as confirmed by FE-SEM results (PT: 71.66 nm (Figure 6a), PTS-0.1: 82.68 nm (Figure 6b), PTS-0.3: 115.8 nm (Figure 6c), PTS-0.5: 165.4 nm (Figure 6d)).

AFM was employed to measure the surface property more precisely, and the results are shown in Figure 7. Before interfacial polymerization, the PES substrate had an average surface roughness of about 6.89 nm due to its porous structure. However, after interfacial polymerization (without SDS), the roughness was measured to be about 1.44 nm. The addition of SDS increased the roughness to 1.53 nm (PTS-0.1) and 3.52 nm (PTS-0.5), depending on its concentration. The result further indicates that SDS significantly enhances the reactivity and thus degree of interface polymerization. These thin-film composite (TFC) hollow fiber membranes were fabricated by varying the SDS concentration, that is, the ratio by weight of SDS to the aqueous solution varied, from 0 to 0.5 wt.%. These measurements confirm that the morphological features of membranes can be significantly affected by the addition of SDS and its concentration. As a surfactant, SDS molecules show different morphological characteristics depending on the presence of the surfactant and the concentration, as the ionic head moves toward the solution, and the non-polar tail tends to move away from the solution [46,47,48,49]. Thus, it enhances the reactivity of interface polymerization.

Next, the surface and cross-section images of the thin-film composite hollow fiber membranes (CMMMs) where mGO concentration was varied, the PEI-TMC-SDS-mGO (PTSM) specimens, were examined, as shown in Figure 6i–l (surface) and Figure 6m–p (cross-section), respectively. These PTSM membranes were fabricated via interfacial polymerization while varying the mGO nanoparticle concentration (i.e., the ratio by weight of mGO to PEI at 0.05, 0.1, 0.25, and 0.35 wt.%) with the SDS concentration fixed at 0.3 wt.%. The specimens were named PTSM-0.05, PTSM-0.1, PTSM-0.25, and PSTM-0.35, respectively, depending on the mGO concentration. Surface roughness and contact angle measurement of TFC hollow fiber membranes were exhibited in Appendix A.

The FE-SEM results showed no significant difference in surface morphology, but the cross-section images revealed that the active layer thickness increased with increasing mGO concentration (102.0 nm to 151.6 nm). This increase in thickness is attributed to the increase in mGO loading in the CMMMs, which is expected to enhance the free space inside the thin-film composite layer. The resultant membranes could enhance the gas separation performance of the CMMMs.

### 3.3. Gas Separation Performance of the Thin-Film Composite Hollow Fiber Membranes, CMMMs

The single gas permeance of the CMMMs membranes was measured using CO_2_ and N_2_ at 25 °C, and the selectivity was obtained by calculating the ratio between CO_2_ permeability to N_2_ permeability. The results are shown in Figure 8.

It was observed that the use of hydrophilic substrate in the IP polymerization dramatically enhanced the gas permeance and selectivity of the resultant interfacially polymerized thin-film hollow fiber membrane. In a previous study, we fabricated membranes of the same type via interfacial polymerization by adding PEI and TMC at the same concentration as in the present study [22]. Surprisingly, the current thin-film hollow fiber membrane specimen (PT), created with the same concentrations of PEI and TMC but different hydrophilic PES substrate incorporated, exhibited CO_2_/N_2_ selectivity about seven times higher than the membranes in the previous study under the same experimental conditions (Appendix A). This increase is attributed to the addition of the hydrophilic polymer substrate, which significantly increased the selectivity of the target gas, CO_2_. This result highlights the importance and significance of the hydrophilic substrates used for interfacial polymerization previously deemed insignificant.

Next, the effect of SDS concentration in the CMMMs on CO_2_ permeance and selectivity was examined. It was observed that CO_2_ permeance increased with the increase in SDS concentration up to a specific limit (0.3%) followed by a decrease at 0.5 % loading (Figure 8a,b), which indicates that the optimum level of SDS surfactant for enhancing CO_2_ separation is 0.3%. The tests were conducted while varying the operating pressure in a range from 0.25 to 2.0 bar. Figure 8a shows how the CO_2_ permeance varied depending on the operating pressure. Higher permeance was observed at a lower operating pressure in all the membrane modules used for CO_2_ separation tests. The PTS-0.3 membrane module, in particular, exhibited the highest CO_2_ permeance of about 44 GPU when the operating pressure was 0.25 bar. However, as the operating pressure was increased above 0.25 bar, the permeance tended to decrease. This result may be attributed to the sorption saturation of the CO_2_-philic functional group of the thin-film composite membrane at high pressure, which is a widely established phenomenon usually observed in facilitated transport membranes.

Among the monomers used in interfacial polymerization, PEI contains many primary, secondary, and tertiary amino groups, and has a strong affinity for CO_2_ [46,47]. As shown in the equations below, primary and secondary amine groups allow facilitated transport via the zwitterion mechanism even in a dry state. Tertiary amines can also perform facilitated transport via Van der Waals force and electrostatic attraction, unlike primary and secondary amines [22,48]. Thus, these fixed amine functional groups can be efficient carriers of CO_2_. In addition, the facilitated transport mechanism, which refers to the increase in CO_2_ selectivity via reversible reactions, can further enhance permeance and selectivity.

Primary amine interaction with CO_2_
First step: RNH_2_ + CO_2_ ↔ RNH_2_^+^COO^−^(3)
Second step: RNH_2_^+^COO^−^ + RNH_2_ ↔ RNHCOO^−^ + RNH_3_^+^(4)

Secondary amine interaction with CO_2_
First step: R_2_NH + CO_2_ ↔ R_2_NH^+^ COO^−^(5)
Second step: R_2_NH^+^ COO^−^ + R_2_NH ↔ R_2_N COO^−^ + R_2_NH_2_^+^(6)

Next, the effect of the SDS concentration on the CO_2_/N_2_ selectivity was observed (Figure 8b). The tendency for increasing SDS concentration to enhance gas permeance can be explained by the fact that, as polar groups accumulate in the polymer structure during interfacial polymerization, the repulsion between polymer chains increases, causing the free volume to increase [49]. As a result, the surface interfacial energy increases from 124.0 to 134.4 mJ/m^2^ as the SDS concentration increases, increasing the specific surface area of the membrane. This increase also indicates that gases can be more actively absorbed and diffused [40]. However, when the concentration exceeded a certain level, the thickness of the coated layers also increases from 102.0 to 151.6 nm due to the increasing concentration of mGO, as shown in the SEM images in Figure 4. Thus, an increase in the coated layer thickness increases the gas permeance barrier, causing the permeance to decrease. Indeed, the experimental results (Figure 8b) also show that the CO_2_/N_2_ selectivity increased from about 23 (0 wt.% of SDS) to 31 (0.1 wt.% of SDS), peaked at 46 (0.3 wt.% of SDS), but then decreased to 37 (0.5 wt.% of SDS). Likewise, the optimal SDS concentration was determined to be 0.3 wt.%. Therefore, when fabricating thin-film composite hollow fiber membranes using mGO, the SDS concentration was fixed at 0.3 wt.%.

The CO_2_ and N_2_ permeance and CO_2_/N_2_ ideal gas selectivity of the thin-film composite hollow fiber membranes (CMMMs) fabricated with mGO concentrations (PSTM) ranging from 0.05 to 0.35 wt.% were measured at room temperature and an operating pressure of 0.25 bar (Figure 8c). As shown in the graph, the permeance of PSTM-0.25 with mGO added was 73 GPU, much higher than that of PTS-0.3 with no GO added, a 66% increase. The CO_2_/N_2_ selectivity increased from 15 to 60.

When the mGO concentration exceeded a certain level, both gas permeance and selectivity started to decrease. This phenomenon can be explained by when the mGO concentration is low (PSTM-0.25), the mGO stacked in the polyamide layers formed via interfacial polymerization and increases the free volume of the thin-film composite layer. Furthermore, the increasing content of mGO also increases the number of ether oxygen and carboxyl groups, which have a strong affinity with CO_2_ and thus enhance the gas permeance [50,51]. However, when the mGO content exceeds a certain level, mGO nanoparticles start to agglomerate and act as a gas diffusion barrier. As shown in the SEM results, this also increases the active layer thickness (up to 151.6 nm), thereby degrading the gas permeance and selectivity.

Figure 8c,d show the CO_2_ and N_2_ permeance and selectivity of the thin-film composite hollow fiber membranes with added mGO and GO. All tests were performed at the same temperature and operating pressure conditions. PTSM-0.1 (0.1 wt.% of mGO with respect to PEI) and the thin-film composite hollow fiber membrane with the same concentration of GO (0.1 wt.% of GO with respect to PEI) were subjected to permeation tests. The results showed that the addition of mGO and GO into IP polymerization facilitates the gas transport significantly, as observed in the PT, PTS, and PTSM membranes, where higher CO_2_ permeance is achieved at lower operating pressure due to the presence of amine-based monomers. At an operating pressure of 0.25 bar, the CO_2_/N_2_ selectivity of the mGO-contained membrane was about 60, and the selectivity of the GO-containing membrane was about 58; the difference was not significant. However, the CO_2_ permeance of mGO and GO was 73 and 53 GPU, respectively, a difference of about 30%. GO is composed of hydroxyl, carboxyl, and epoxy groups containing oxygen. The material is easy to synthesize and features various surface chemical reactions. It is widely used to fabricate stacks and stacked laminates [52]. Stacked laminates formed by mGO with a defect structure inside a membrane are less resistant to permeating gases than GO-based stacked laminates. Therefore, the nanosized particles, which allow gas molecules to permeate through graphene single layers, lowered energy barriers for gas permeation and facilitated gas molecule transport, increasing permeation performance [53]. Summary of CO_2_ permeance and CO_2_/N_2_ selectivity MMMs membranes was shown in Appendix A.

## 4. Conclusions

Thin-film composite hollow fiber membranes combined with hydrophilic hollow fiber support and mGO nanoparticles were fabricated and analyzed. First, a hollow fiber membrane substrate was fabricated via phase inversion using the spinning process by adding poly(ethylene glycol) (PEG) and dopamine (DOPA) to the fabricated poly(ether sulfone) (PES) substrate. Then, the obtained hollow fiber membrane substrate was coated via interfacial polymerization using the interaction between polyethylenimine (PEI) as an aqueous monomer and 1,3,5-benzenetricarbonyl trichloride as an organic monomer.

The addition of the hydrophilic hollow fiber membrane substrate was found to increase the CO_2_ permselectivity 7-fold compared to the pristine PES hollow fiber substrate. This result highlights the importance and significance of membrane substrates.

Interfacially polymerized thin layers were formed in the presence of sodium dodecyl sulfate (SDS) as a surfactant. The IP technique using SDS has not yet been reported for gas separation application. Therefore, the surface interfacial energy, thickness, and gas permeation properties of the coated layers were examined with respect to the SDS concentration. The optimal SDS concentration was determined based on the results. Porous graphene oxide (mGO) was also synthesized and used as an additive for the thin-film composite hollow fiber membranes. The results of gas permeance tests using CO_2_ and N_2_ confirmed that the membrane containing mGO nanoparticles achieved CO_2_ permeance of 73 GPU, a 66% increase over the performance of PTS-0.3 with no mGO added.

The current approach successfully demonstrates the importance of the hydrophilic substrate for interfacial polymerization on the hollow fiber membrane. It also clearly displays that the gas separation performance of the thin-film composite hollow fiber membrane can be enhanced effectively by using SDS for the IP process in the thin-film composite layer, as well as by incorporating mGO in the thin-film composite hollow fiber membrane.

## Figures and Tables

**Figure 1 membranes-11-00650-f001:**
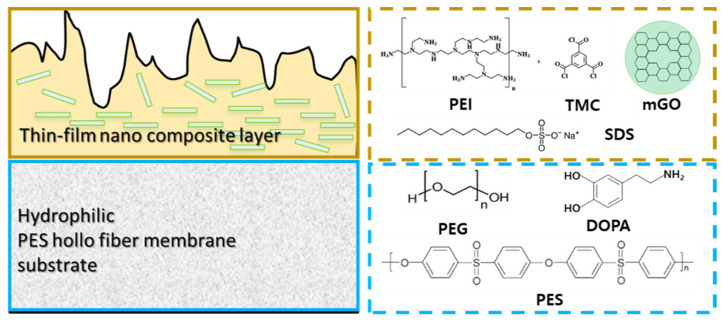
Schematic diagram of an mGO-thin-film composite hollow fiber membrane.

**Figure 2 membranes-11-00650-f002:**
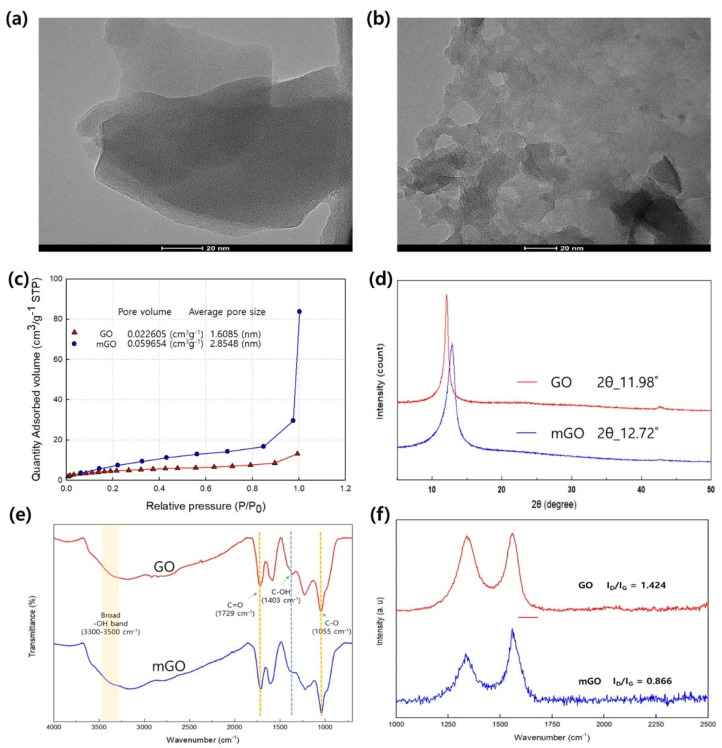
Characterization of GO nanoparticles. (**a**) TEM image of GO (before synthesis), (**b**) TEM image of mGO (after synthesis), (**c**) BET analysis results for GO and mGO, (**d**) XRD analysis results, (**e**) FT-IR analysis results for GO and mGO, and (**f**) Raman spectroscopy analysis results for GO and mGO.

**Figure 3 membranes-11-00650-f003:**
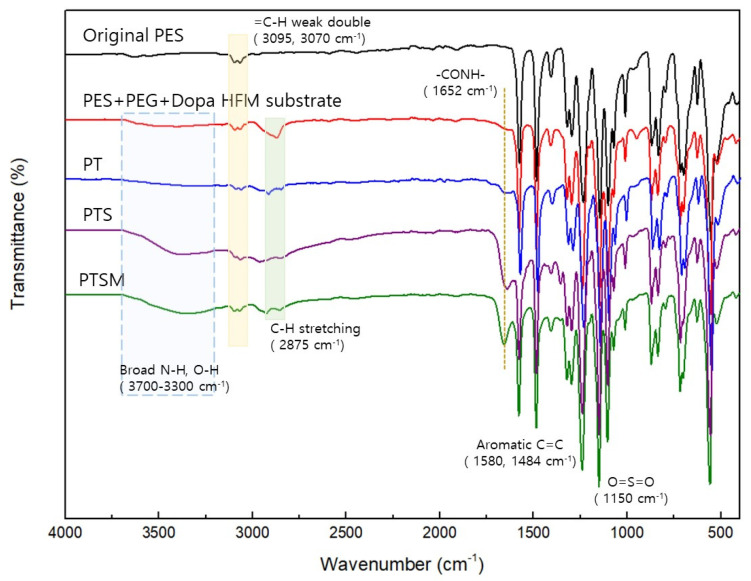
ATR-FTIR spectra of the thin-film composite hollow fiber membranes.

**Figure 4 membranes-11-00650-f004:**
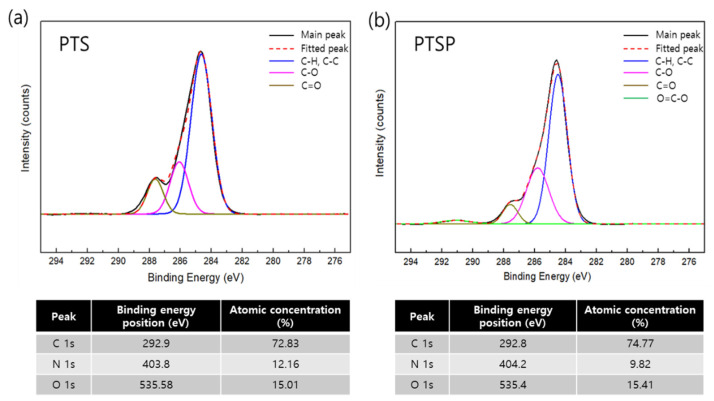
XPS graphs (C1s) and atomic concentration for (**a**) PTS and (**b**) PTSM membranes.

**Figure 5 membranes-11-00650-f005:**
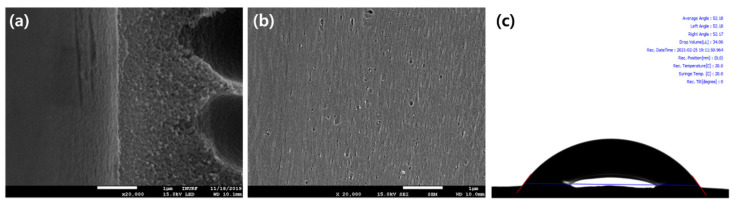
FE-SEM of the hydrophilic hollow fiber membrane. (**a**) cross-section image (20.0 K), (**b**) surface image, and (**c**) contact angle image of the hydrophilic substrate.

**Figure 6 membranes-11-00650-f006:**
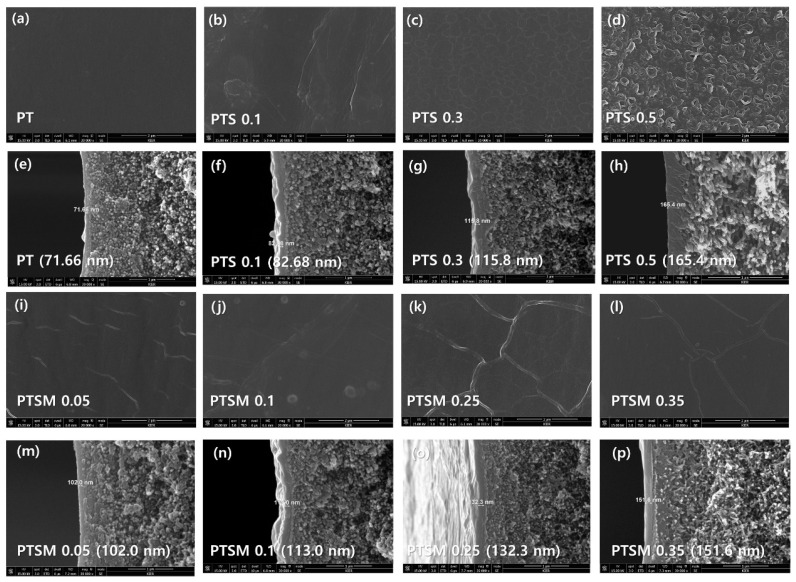
FE-SEM images of the thin-film composite hollow fiber membranes. Surface image (20.0 K amplification) of (**a**) PT, (**b**) PTS-0.1, (**c**) PTS-3.0, (**d**) PTS-0.5 CMMMs; cross-section image (30.0 K amplification) of (**e**) PT, (**f**) PTS-0.1, (**g**) PTS-3.0, (**h**) PTS-0.5 CMMMs; surface image (20.0 K amplification) of (**i**) PTSMP-0.05, (**j**) PTSM-0.1, (**k**) PTSM-0.25, (**l**) PTSM-0.35 CMMMs, cross-section image (30.0 K amplification) of (**m**) PTSP-0.05, (**n**) PTSM-0.1, (**o**) PTSM-0.25, and (**p**) PTSP-0.35 CMMMs.

**Figure 7 membranes-11-00650-f007:**
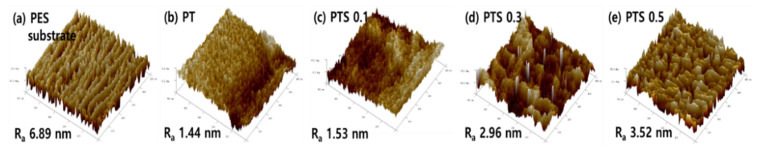
Three-dimensional AFM images of the thin-film composite membranes: (**a**) PES hollow fiber substrate, (**b**) PT, (**c**) PTS 0.1, (**d**) PTS 0.3, and (**e**) PTS 0.5.

**Figure 8 membranes-11-00650-f008:**
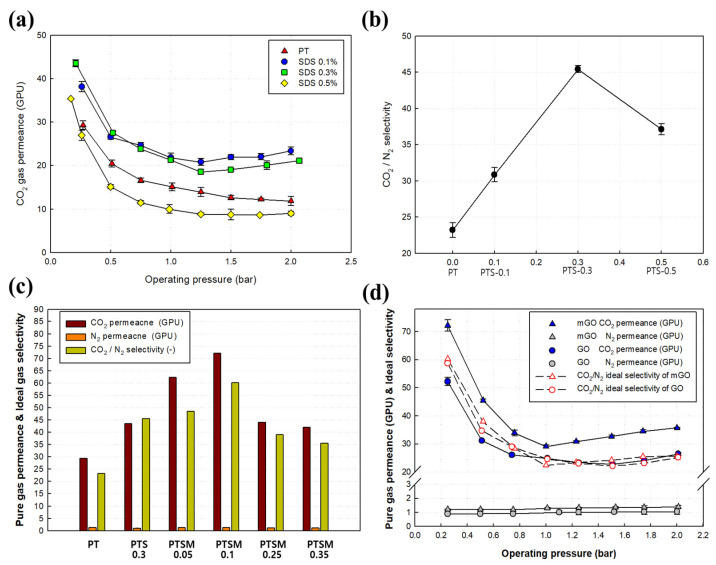
The impact of pressure, SDS, and GO concentration on CO_2_ and N_2_ permeance and selectivity. (**a**) Effects of the operating pressure and membrane type on CO_2_ permeance. (**b**) Ideal selectivity of the PT and PTS TFC hollow fiber membranes, (**c**) Thin-film composite hollow fiber membrane permeance and ideal selectivity (**d**) Comparison of CO_2_ and N_2_ permeance and ideal selectivity between mGO (PTSP 0.1) and the GO thin-film composite hollow fiber membrane.

**Table 1 membranes-11-00650-t001:** Composition of the dope solution and spinning conditions for the preparation of PES hollow fiber membranes.

**Composition (wt.%)**	
PES	19.15
NMP	35.1
PEG 6000	15.32
Dopamine	0.38
Ammonium persulfate	0.001
**Spinning Conditions**	
Air gap	0 cm
Spinneret ID/OD	0.15/0.9 mm
Internal coagulant	DI water

**Table 2 membranes-11-00650-t002:** Compositions of GO-thin-film composite hollow fiber membranes.

Entry	Material Compositions			mGO Concentration Relative to PEI (wt.%)
	PEI (Molar Ratio)	TMC (Molar Ratio)	SDS (wt.%)	
PT (PEI-TMC)	2	1	0	0
PTS-0.1 (PEI-TMC-SDS)	2	1	0.1	0
PTS-0.3	2	1	0.3	0
PTS-0.5	2	1	0.5	0
PTSM-0.05 (PEI-TMC-SDS-mGO)	2	1	0.3	0.05
PTSM-0.1	2	1	0.3	0.1
PTSM-0.25	2	1	0.3	0.25
PTSM-0.35	2	1	0.3	0.35

**Table 3 membranes-11-00650-t003:** Experimental conditions for single gas permeance.

Experimental Conditions	
Operating pressure	0.25–2.0 bar
Operating temperature	25 °C
Gas composition	CO_2_, N_2_, 99.99%

## Data Availability

Not applicable.

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
