# Peer review of "Modified Graphene Oxide-Incorporated Thin-Film Composite Hollow Fiber Membranes through Interface Polymerization on Hydrophilic Substrate for CO2 Separation"

_membranes, 2021, doi:10.3390/membranes11090650_

Round 1

Reviewer 1 Report

The manuscript discussed a thin-film composite hollow fiber membrane on PES substrate for CO2 separation. It is not a noval topic. This work lacks creativity and theoretical depth.

  1. The title is too general and unclear. It had to be reviesed to a clear one.
  2. Section 3.1 is about GO pretreatment with base solution. It is called as modified GO in the manuscript. How does the GO improve gas permeation? What is the relationship between ID/IG ratio and number of oxygen-containing functional groups?
  3. How is the active layer microstructure of PTSM? Does the thinner layer(151.6nm) take great effect on the CO2 permeate flux?

Author Response

  1. Response to reviewer 1’s comments:

Most critical issues were resolved, but there are still a couple of inaccuracies in the manuscript which need attention.

[Answers to general comments]

  • Thank you for your valuable comments.
  • We have responded to all the comments raised by the reviewers and have revised the manuscript accordingly.
  • Answers for each comment are given as follows:

Q1) The title is too general and unclear. It had to be revised to a clear one.

[Answers to comment 1]

  • We have revised the title as follows:

Original: Thin-film Composite Hollow Fiber Membranes for CO2 Separation: Roles of Modified Graphene Oxide, Surfactant, and Hydrophilic Substrate

Revised: “Modified Graphene Oxide-Incorporated Thin-Film Composite Hollow Fiber Membranes through Interface Polymerization on Hydrophilic Substrate for CO2 Separation”

Q2) Section 3.1 is about GO pretreatment with base solution. It is called as modified GO in the manuscript. How does the GO improve gas permeation? What is the relationship between ID/IG ratio and number of oxygen-containing functional groups?.

[Answers to comment 2]

  • We described the effect of mGO in Part 3.3 as follows.

The mGO stacked in the polyamide layers formed via interfacial polymerization and increases the free volume of the thin-film composite layer. Furthermore, the increased content of mGO also increases the number of ether oxygen and carboxyl groups, which have a strong affinity with CO2 and thus enhance the gas permeance [50,51].

  • Regarding the ID/IG ratio, we infer that we are in a region in which ID/IG increases with increased disorder, and we adopted this parameter to interpret our results. King et al. have suggested that the peak position difference between D and G is directly related to the oxygen content in GO. This may be attributed to the oxidation of unstable functional groups and has also been confirmed by ID/IG spectral analysis with decreasing oxygen content. Also, please see the following references, Rep. 2016, 6,19491 (doi10.1038/srep19491) and https://doi.org/10.3390/c7020048

Q3) How is the active layer microstructure of PTSM? Does the thinner layer(151.6nm) take a great effect on the CO2 permeate flux?

[Answers to comment 3]

  • The microstructure of the active layer has been explored in detail in the manuscript using XPS, SEM, and AFM analysis.
  • We used a porous PES substrate to achieve selectivity with high permeance. In addition, to improve selectivity, a polyimide layer with a dense structure was applied using a thin-film coating method. As the reviewer notes, the thickness of the coating layer affects the permeance and selectivity: the thicker it is, the lower the permeance is, due to the membrane resistance, as mentioned in the text. Therefore, after coating, the structure was a hollow fiber membrane with an asymmetric structure e, and research was conducted to determine optimal permeability and selectivity.

Reviewer 2 Report

In this paper, the authors designed fabricated thin-film composite mixed matrix membranes (CMMMs) using interfacial polymerization to achieve high permeance and selectivity for CO2 separation. I agree that this topic is very relevant for materials and environmental fields. Also, I do see good importance of this paper for the readers. In my opinion the novelty of this work is satisfactory. This work is well written and has enough characterization data compared to other previous works reported in literature.

Some points need to be clarified before I recommend its publication.

 Specific comments  

1 – In introduction, clearly build your research hypothesis (straightforward question that is answerable by yes or no). I am not sure this is clear in the manuscript. 

2 – Please include more informations about methods, to be reproducible (ex: XRD, ATR-FTIR, X ray photoelectron spectroscopy, etc) 

3 – I page 9, ‘’In the thin-film composite hollow fiber membranes after interfacial polymerization, a -CONH- peak was observed at 1652 cm-1, attributed to the polyamide layers formed via  crosslinking between the PEI and TMC.’’ This sentence should be properly referenced. 

4 – For the XPS results, it would be very interestin if a Table with the quantitative information of the main elements (C,O, H and N) would be added. This would give interesting information about how hydrophilic the sample is. 

5 – What is the role oif the porosity of the membranes on its efficiency ? if there is any. 

6 – In page 11…. ‘’These thin-film composite (TFC) hollow fiber membranes were fabricated by varying the SDS concentration, that is, the  ratio by weight of SDS to the aqueous solution varied, from 0 to 0.5 wt.%. These measurements confirm that the morphological features of membranes can be significantly affected by the addition of SDS and its concentration…’’.Affected how? further explanation should be added. 

7 – In page 11…. ‘’This increase in thickness may be attributed 403 to the increase of free space inside the thin-film composite layer, which is highly desired 404 for enhancing the gas separation performance of the resultant CMMMs.’’ Can this be observed by SEM analysis? 

8 – In page 12…. ‘’This increase is attributed to the addition of the hydrophilic polymer substrate, which significantly increased the selectivity of the target gas, CO2. This result highlights the importance and significance of the hydrophilic substrates used for interfa- 429 cial polymerization previously deemed insignificant.’’ I believe this sentence should be properly referenced. 

9 – Please cite at least 5 recent papers (2020-2021) from Membranes.

Author Response

  1. Response to reviewer 2’s comments:

In this paper, the authors designed fabricated thin-film composite mixed matrix membranes (CMMMs) using interfacial polymerization to achieve high permeance and selectivity for CO2 separation. I agree that this topic is very relevant for materials and environmental fields. Also, I do see good importance of this paper for the readers. In my opinion the novelty of this work is satisfactory. This work is well written and has enough characterization data compared to other previous works reported in literature.

Some points need to be clarified before I recommend its publication.

[Answers to general comments]

  • Thank you for your valuable comments.
  • We have responded to all the comments raised by the reviewers and have revised the manuscript accordingly.

Q1) In introduction, clearly build your research hypothesis (straightforward question that is answerable by yes or no). I am not sure this is clear in the manuscript.

[Answers to comment 1]

  • We strongly believe that our research hypothesis has been explained in the introduction.

Q2) Please include more informations about methods, to be reproducible (ex: XRD, ATR-FTIR, X ray photoelectron spectroscopy, etc)

[Answers to comment 2]

  • We have revised the experimental section as follows:

Revised: “The characteristics of the hollow fiber membrane substrate fabricated by the spinning process, the synthesized mGO, and the thin-film composite hollow fiber membranes were analyzed as follows. An X-ray diffractometer (XRD; Rigaku SmartLab) was used to analyze the changes in the d-spacing of the GO and mGO within the range of 5° ≤ 2θ ≤ 40°. The morphological analysis was performed using a transmission electron microscope (TEM; Talos F200X). Raman spectroscopy (Renishaw inVia Qontor) was used to measure the D/G bandgap and determine whether the intended phases had been synthesized. X-ray photoelectron spectroscopy (XPS; PHI 5000 Versaprobe II) and attenuated total reflectance Fourier transform infrared spectroscopy (ATR-FTIR) were also employed to analyze the wavenumber range 600–4000 cm−1. The cross-section and surface of the hollow fiber membranes and the substrate were observed and analyzed using a field emission-scanning electron microscope (FE-SEM; JEOL/JSM_7800F). In order to obtain an image of the cross-section of the hollow fiber membrane, it was cut using liquid nitrogen after it had been wetted with DI water. The surface roughness of the membranes was examined using an atomic force microscope in non-contact tapping mode (AFM; Bruker, MULTIMODE-8-AM). The surface hydrophilicity of the fabricated hollow fiber membranes was analyzed by measuring the contact angles (Phoenix 300 Plus, SEO). The surface hydrophilicity was measured at room temperature by the sessile-drop method using 3 μL of DI water. Each sample was measured at 3 or more locations, and the average was used.

Q3) page 9, “In the thin-film composite hollow fiber membranes after interfacial polymerization, a -CONH- peak was observed at 1652 cm-1, attributed to the polyamide layers formed via crosslinking between the PEI and TMC.” This sentence should be properly referenced.

[Answers to comment 3]

We added reference 22 in the experimental section:

“In the thin-film composite hollow fiber membranes after interfacial polymerization, a CONH peak was observed at 1652 cm−1, attributed to the polyamide layers formed via crosslinking between the PEI and TMC [22].”

Q4) For the XPS results, it would be very interesting if a Table with the quantitative information of the main elements (C,O, H and N) would be added. This would give interesting information about how hydrophilic the sample is.

[Answers to comment 4]

  • Upon reviewer request, we have added a table below the XPS profile as follows:
    • The proportions of carbon, oxygen, and nitrogen in the polyamide of the TFN-based hollow fiber membrane are also shown. Since XPS can penetrate only to a depth of 5 nm inside the surface, the presence of nanoparticles in the polyamide layer cannot be accurately confirmed. Also, the hydrogen composition cannot be determined through XPS. And the hydrophilic part has been addressed using FTIR spectroscopy.

Q5) What is the role of the porosity of the membranes on its efficiency? if there is any.

[Answers to comment 5]

  • Porous membranes exhibit relatively high levels of flux compared to non-porous membranes. However, they also usually have low selectivity values. Therefore, by using such a porous support, a hollow fiber membrane in the form of a composite membrane was manufactured through interfacial polymerization to maintain high permeability and overcome the low selectivity value.

Q6) In page 11…. “These thin-film composite (TFC) hollow fiber membranes were fabricated by varying the SDS concentration, that is, the ratio by weight of SDS to the aqueous solution varied, from 0 to 0.5 wt.%. These measurements confirm that the morphological features of membranes can be significantly affected by the addition of SDS and its concentration…” Affected how? further explanation should be added.

[Answers to comment 6]

  • We have added the following sentences to the revised manuscript, in addition to the discussion in section 3.3.
  • As a surfactant, SDS molecules show different morphological characteristics depending on the presence of the surfactant and the concentration because the ionic head moves toward the solution, and the non-polar tail tends to move away from the solution [49]. Thus, it enhances the reactivity of interface polymerization.

Q7) This increase in thickness may be attributed 403 to the increase of free space inside the thin-film composite layer, which is highly desired 404 for enhancing the gas separation performance of the resultant CMMMs.” Can this be observed by SEM analysis?

[Answers to comment 7]

  • We apologize for our mistakes. Indeed the free space cannot be analyzed by SEM. However, the increased thickness observed in the SEM image could be evidence for mGO loading. We revised the manuscript as follows:

-Revised: “This increase in thickness is attributed to the increase of mGO loading in the CMMMs, which is expected to enhance the free space inside the thin-film composite layer. The resultant membranes could enhance the gas separation performance of the CMMMs.”

Q8) In page 12…. “This increase is attributed to the addition of the hydrophilic polymer substrate, which significantly increased the selectivity of the target gas, CO2. This result highlights the importance and significance of the hydrophilic substrates used for interfa- 429 cial polymerization previously deemed insignificant.” I believe this sentence should be properly referenced.

[Answers to comment 8]

  • To the best of our knowledge, this is the first investigation of this hydrophilic surface effect. We could not find any appropriate references.
  • The hydrophilic surface improves the CO2 selectivity and permeability. First, when the aqueous solution used during interfacial polymerization meets a hydrophilic surface, it spreads evenly, forming an amide layer with a dense structure on the surface. This was verified through contact angle measurements and SEM imaging. The hydrophilic support is affected by solubility diffusion in CO2

Q9) Please cite at least 5 recent papers (2020-2021) from Membranes.

[Answers to comment 9]

  • Upon the reviewer’s request, we have added the following references in the supporting information.

[S4] C. Regmi, S. Ashtiani, CeO2 -Blended Cellulose Triacetate Mixed-Matrix Membranes for Selective CO2 Separation, Membranes, 11(2021), pp.1-21, 10.3390/membranes11080632.

[S8] K. Papchenko, G. Risaliti, M. Ferroni, M. Christian, M.G. De Angelis, An analysis of the effect of ZIF-8 addition on the separation properties of polysulfone at various temperatures, Membranes (Basel).11(6) (2021), pp.427, 10.3390/membranes11060427

[S14] S. Ashtiani, M. Khoshnamvand, C. Regmi, K. Friess, Interfacial design of mixed matrix membranes via grafting PVA on UiO-66-NH2 to enhance the gas separation performance, Membranes (Basel). 11(6) (2021), pp.419, 10.3390/membranes11060419.

[S15] C.Y. Chuah, J. Lee, J. Song, T.H. Bae, Co2/n2 separation properties of polyimide-based mixed-matrix membranes comprising UiO-66 with various functionalities, Membranes (Basel). 10(7) (2020), pp.154, 10.3390/membranes10070154.

[S16] P.H. Tang, P.B. So, W.H. Li, Z.Y. Hui, C.C. Hu, C.H. Lin, Carbon dioxide enrichment pebax/mof composite membrane for CO2 separation, Membranes (Basel). 11(6) (2021), pp.404, 10.3390/membranes11060404.

Reviewer 3 Report

The following suggestions must be addressed before acceptance.

  1. According to Figure 1, there is a self-assembled organisation of GO in the membrane structure, but there is no analysis method presented to prove that. If you can't prove that the GO is highly organised and parallel in the structure of membrane, please re-drawn the figure.
  2. Line 262 - authors measure the sorption capacity of mGO in comparison with GO. This analysis is performed based on total porosity of analysed material. Where is the porosity of mGO and GO? Please revise and be more explicite. If you have a bundle of mGO, then you are very far from an exfoliated structure of filler (see previous comment)
  3. In Figure 2 - FT-IR of GO and mGO, there are no clear evidences of modifications, basically the two spectra being identically. Since the authors have access to XPS, I strongly recommend a hi-res XPS on C and O for GO and mGO.
  4. From Figure 8, the error bars are missing, thus the reproducibility of synthesised materials being not provided.

Author Response

  1. Response to reviewer 3’s comments:

Q1) According to Figure 1, there is a self-assembled organisation of GO in the membrane structure, but there is no analysis method presented to prove that. If you cant prove that the GO is highly organised and parallel in the structure of membrane, please re-drawn the figure.

[Answers to comment 1]

  • Thank you for your valuable comments.
  • We have responded to all the comments raised by the reviewers and have revised the manuscript accordingly.
  • Upon reviewer request, we have revised the manuscript as follows:
  • Revised:

Q2 Line 262 - authors measure the sorption capacity of mGO in comparison with GO. This analysis is performed based on total porosity of analysed material. Where is the porosity of mGO and GO? Please revise and be more explicite. If you have a bundle of mGO, then you are very far from an exfoliated structure of filler (see previous comment)

[Answers to comment 2]

  • We agree with the reviewer that the adsorption isotherm does not reflect the porous nature of GO and mGO. That is the reason we have not claimed that our material is a porous GO. Instead, we defined it as a modified GO, consistent with our TEM, XRD, and Raman analysis. Moreover, the sorption isotherm and pore size distribution curves also indicate the modification in GO.
  • We have revised our manuscript as follows:

Revised: “The adsorption isotherm indicates that the amount of N2 sorption capacity increased to about double in the modified GO. In addition, the pore size distribution curve also indicates that the pore size increased from 19.8 nm (for GO) to 28.4 nm (for mGO) (Figure S3). These results indicate that the GO has been successfully modified in our experiments.”

Figure S3. Comparison between (a) GO and (b) mGO pore size distribution analysis.

  • Figure S3 has been added to the manuscript. All other figure numbers have been revised accordingly throughout the manuscript.

Q3) In Figure 2 - FT-IR of GO and mGO, there are no clear evidences of modifications, basically the two spectra being identically. Since the authors have access to XPS, I strongly recommend a hi-res XPS on C and O for GO and mGO.

[Answers to comment 3]

  • We are sorry that we cannot provide XPS data within this short period. However, we characterized the modified GO using TEM, BET, pore size distribution, and XRD to prove the modification of GO. All these methods indicate that GO has been modified as the surface morphology, sorption amount, pore size, and d-spacing of mGO are different from GO.
  • Please see Figure S3 and Figure 2 (1-d).
  • Also, please see the answer to question number 2.

Q4) From Figure 8, the error bars are missing, thus the reproducibility of synthesised materials being not provided.

[Answers to comment 4]

  • Upon reviewer request, we have added the error bars in the figures and revised the manuscript as follows:

Revised:

Round 2

Reviewer 1 Report

The authors have revised their manucscript according to the comment. There are some problems.

1)The figures quanlity(dpi may be low) should be improved.

2)Some writing mistakes should be corrected. Subscript problem of CO2 and N2. J. Memb. Sci. should be J. Membr. Sci.. others.